# Proteomics Reveals the Potential Protective Mechanism of Hydrogen Sulfide on Retinal Ganglion Cells in an Ischemia/Reperfusion Injury Animal Model

**DOI:** 10.3390/ph13090213

**Published:** 2020-08-27

**Authors:** Hanhan Liu, Natarajan Perumal, Caroline Manicam, Karl Mercieca, Verena Prokosch

**Affiliations:** 1Experimental and Translational Ophthalmology, University Medical Centre of the Johannes Gutenberg University Mainz, 55131 Mainz, Germany; hanhan_liu@163.com (H.L.); nperumal@eye-research.org (N.P.); caroline.manicam@unimedizin-mainz.de (C.M.); 2Royal Eye Hospital, School of Medicine, University of Manchester, Manchester M13 9WH, UK; doctormercieca@yahoo.com; 3Department of Ophthalmology, University Medical Centre of the Johannes Gutenberg University Mainz, 55131 Mainz, Germany

**Keywords:** hydrogen sulfide, glaucoma, neuronal apoptosis, label-free mass spectrometry, mitochondrial function, signalling pathways

## Abstract

Glaucoma is the leading cause of irreversible blindness and is characterized by progressive retinal ganglion cell (RGC) degeneration. Hydrogen sulfide (H_2_S) is a potent neurotransmitter and has been proven to protect RGCs against glaucomatous injury in vitro and in vivo. This study is to provide an overall insight of H_2_S’s role in glaucoma pathophysiology. Ischemia-reperfusion injury (I/R) was induced in Sprague-Dawley rats (*n* = 12) by elevating intraocular pressure to 55 mmHg for 60 min. Six of the animals received intravitreal injection of H_2_S precursor prior to the procedure and the retina was harvested 24 h later. Contralateral eyes were assigned as control. RGCs were quantified and compared within the groups. Retinal proteins were analyzed via label-free mass spectrometry based quantitative proteomics approach. The pathways of the differentially expressed proteins were identified by ingenuity pathway analysis (IPA). H_2_S significantly improved RGC survival against I/R in vivo (*p* < 0.001). In total 1115 proteins were identified, 18 key proteins were significantly differentially expressed due to I/R and restored by H_2_S. Another 11 proteins were differentially expressed following H_2_S. IPA revealed a significant H_2_S-mediated activation of pathways related to mitochondrial function, iron homeostasis and vasodilation. This study provides first evidence of the complex role that H_2_S plays in protecting RGC against I/R.

## 1. Introduction

Glaucoma is one of the leading causes of irreversible blindness worldwide [1] and is characterized by progressive optic nerve and retinal ganglion cell (RGC) degeneration [2]. Although elevated intraocular pressure (IOP) is the main risk factor and mainstay of treatment, simply lowering IOP may fail to halt the disease progression in many patients despite delaying RGC death. Novel strategies which halt RGC loss are desperately needed to prolong visual function in glaucoma [3].

Along with carbon monoxide (CO) and nitric oxide (NO), hydrogen sulfide (H_2_S), has been recognized as a third endogenous gaseous signaling molecule [4]. The importance of NO in biology and medicine was highlighted in 1998 when the Nobel Prize was awarded in Physiology and Medicine to Robert Furchgott, Louis Ignarro and Ferid Murad for their pioneering work on the role of NO in the nervous, cardiovascular and immune systems [5]. In the same time period, CO has also been recognized as putative neurotransmitters [6]. They are both involved in several important aspects of neuronal function. In physiological condition, NO formed in the eye is beneficial in preventing the genesis of glaucoma or halting its progress; however, excessive NO would become pathogenic in the progress of glaucomatous optic neuropathy [7]. In vitro and in vivo studies have shown evidence of NO’s involvement in retinal degeneration, the regulation of IOP and the production of aqueous humor [8,9,10]. Drugs inhibiting NO formation showed protective effects in the retina against elevated IOP in glaucoma animal models [11,12,13]. On the other hand, The CO production system in the retina has been highlighted as a protective mechanism, its protecting role is demonstrated in the regulation of Ischemia-reperfusion (I/R) -induced damage, protecting retinal ganglion cells against oxidative damage, increases retinal and choroidal blood flows and downregulating inflammatory response in the anterior segment of the eye [14,15,16]. Only recently, has H_2_S been recognized as a physiological gasotransmitter of comparable importance to NO and CO [17]. It is produced via cysteine catabolism by the cytoplasmic enzymes, cystathionine-β-synthase (CBS) and cystathionine-γ-lyase (CSE), and also via 3-mercaptopyruvate catabolism by 3-mercaptopyruvate sulfurtransferase (3-MST) [18]. H_2_S has recently been considered an endogenous gasotransmitter with protective potential at low concentrations. Alteration of endogenous H_2_S levels in the retina is also linked to different pathological conditions, and its exogenous donors have been shown to exhibit potential in protecting retinal ganglion cells against insults such as diabetic retinopathy, ischemia-reperfusion injury and N-methyl-D-aspartic acid (NMDA)-induced excitatory neurotoxicity [16,19,20]. We showed in our previous study that endogenous H_2_S synthesis via 3-MST is increased in a glaucoma animal model after seven weeks of IOP elevation. Furthermore, we documented that GYY4137, a slow-release H_2_S donor, effectively protected RGC against different glaucomatous injuries in vitro and in vivo, in a dose-dependent manner [21].

The neuroprotective effect of H_2_S was partly attributed to its capability of vasorelaxation, anti-oxidative stress, neuroendocrine regulation and inflammation suppression [22,23,24], but our current understanding of the mechanism behind RGC apoptosis and protective properties of H_2_S is far from comprehensive.

In this study we examined the potential protective mechanisms activated by H_2_S in a glaucoma animal model. In this work, we used mass spectrometry-based proteomics to elucidate how protein expression changes at the cellular level, in H_2_S-treated and H_2_S-untreated retinae of Sprague-Dawley rats that underwent acute elevated IOP (ischemia-reperfusion injury), and to provide a more precise direction for further studies.

## 2. Results

### 2.1. Pretreatment of H_2_S Improves RGC Survival against Ischemia-Reperfusion Injury

IOP of the animals was elevated to 55 mmHg for 60 min to induce regional ischemic injury, followed by 24 h of reperfusion. I/R injury resulted in significant reduction in the number of RGC in the operated eyes (1101.7 ± 116.4 RGC/mm^2^) compared to the contralateral control (1310.6 ± 236.4 RGC/mm^2^). While injection of the H_2_S precursor, GYY4137, prior to I/R injury significantly improved RGC survival (1295.4 ± 136.1 RGC/mm^2^). Total number of RGC between control and H_2_S treated group is not significantly different (Figure 1) (*** *p* < 0.001, ** *p* < 0.01, *n* = 6 per group, means ± SD).

### 2.2. Proteomic Profiling

MS-based discovery proteomics was used to identify and distinguish proteome profiles of rat retina undergoing I/R injury with or without pre-treatment of H_2_S. In total 1115 retinal proteins were identified with a false discovery rate (FDR) of 1% (the complete list of proteins can be found as Appendix A “Total proteins”). Ischemia/reperfusion injury with (1046 proteins) or without H_2_S pretreatment (1015 proteins) resulted in the expression of a slightly lower number of proteins than the control group (CTRL; 1070 proteins). All three groups showed an overlap of 963 proteins (Figure 2B). Furthermore, 34 proteins were exclusively expressed in the control, 14 proteins in I/R group and H_2_S group.

### 2.3. Differential Expression of Retinal Proteins between I/R and H_2_S Groups

A total number of 79 proteins were found to be significantly differentially expressed (*p* < 0.05) in the retinal samples of both I/R and H_2_S groups compared to the control group. A slightly higher correlation was observed between the H_2_S vs. CTRL group compared with I/R vs. CTRL group, which indicates high similarities between the proteome of control and H_2_S pre-treated groups. A heat map with unsupervised hierarchical clustering of retina proteome in I/R, H_2_S and CTRL groups shows the segregation of identified proteins into various clusters (Figure 2C, also can be found in Appendix A).

Forty-eight proteins were found to be differentially expressed (*p* < 0.05) in the retina samples of the I/R group compared to the CTRL group (listed in Table 1, also see Figure 3A). The five most significantly down-regulated proteins are 116 kDa U5 small nuclear ribonucleoprotein component (*EFTUD2*), E3 ubiquitin-protein ligase NEDD4 (*Nedd4*), adenylyl cyclase-associated protein 1 (*CAP1*), mucin-19 (*MUC19*) and hydroxymethylglutaryl-CoA lyase, mitochondrial (*Hmgcl*). The five most significantly up-regulated proteins are vesicular inhibitory amino acid transporter(*SLC32A1*), inorganic pyrophosphatase (*PPA1*), AP-2 complex subunit mu (*AP2M1*), mannose-P-dolichol utilization defect 1 protein (*MPDU1*) and NADH-cytochrome b5 reductase 3(*Cyb5r3*).

Closer examination of the protein levels within each of the dominant clusters showed two major clusters of proteins that were involved in the mechanism underlying the neuroprotective properties of H_2_S against I/R.

First, among 48 proteins, which were differentially expressed in the retina samples of I/R group compared to CTRL group, a cluster of 18 proteins can be restored to near normal or normal level by H_2_S pretreatment, which comprised 22.78% of the total differentially expressed proteins due to I/R injury (Table 2, Figure 3B,C). Proteins were classified by PANTHER classification system into six categories, which are cytoskeletal protein, membrane traffic protein, metabolite interconversion enzyme, nucleic acid binding protein, protein modifying enzyme and translational protein.

The five proteins, which are most significantly up-regulated by H_2_S pretreatment compared to I/R group, are CaM kinase-like vesicle-associated protein (*CAMKV*), trifunctional enzyme subunit alpha (*Hadha*), solute carrier family 2, facilitated glucose transporter member 1(Slc2a1), Hmgcl and SURP and G-patch domain-containing protein 2 (*SUGP2*). Furthermore, the proteins that were significantly down-regulated by H_2_S, included mannose-P-dolichol utilization defect 1 protein (*MPDU1*), tripeptidyl-peptidase 2 (*Tpp2*), NADH-cytochrome b5 reductase 3(*Cyb5r3*), eukaryotic translation initiation factor 3 subunit E (*Eif3e*) and 26S proteasome non-ATPase regulatory subunit 13 (*PSMD13*).

The second cluster was characterized by the expression of certain proteins, which were only differentially expressed due to pre-treatment of H_2_S (Table 3, Figure 3D–E). Proteins were classified by PANTHER classification system into five categories, calcium-binding protein, cytoskeletal protein, metabolite interconversion enzyme, nucleic acid binding protein and translational protein. Pre-treatment of H_2_S significantly increased the abundance of glutaredoxin-3(*Glrx3*), dynein light chain roadblock-type 1(*Dynlrb1*) and probable ATP-dependent RNA helicase DDX5(*DDX5*), and significantly decreased the abundance of small nuclear ribonucleoprotein G(*SNRPG*), annexin A6(*Anxa6*) and keratin, type II cytoskeletal 1(*KRT1*).

### 2.4. Pathway Analysis of the Differentially Expressed Retinal Proteins in I/R and H_2_S Groups

To better understand the roles of H_2_S in I/R injury, we utilized the Ingenuity Pathway Analysis (IPA) software to identify the protein-protein interaction (PPI) networks and canonical pathways of the differentially expressed proteins. The most significantly affected pathways due to I/R comprised ketogenesis are, formaldehyde oxidation II, phototransduction pathway, anandamide degradation and heme degradation (Table 4). Amongst which, pathways such as ketogenesis, gamma-aminobutyric acid (GABA) receptor signaling, leucine degradation I and ketolysis, are significantly restored following H_2_S pre-treatment.

Compared to the CTRL group, administration of H_2_S also significantly modulated pathways, such as NER pathway, oxidative phosphorylation and estrogen receptor signaling (Table 5).

To further explore the protein–protein interaction (PPI) networks of the clusters of differentially expressed proteins, functional pathway enrichment was analyzed in the I/R group, H_2_S-restored proteins and H_2_S-modulated proteins. The global view of the interactions between proteins that were differentially regulated in the I/R group is shown in Figure 4. The proteins with the most interactions were EFTUD2, AP2M1, Hadha, Heat shock 70 kDa protein 1-like (HSPA1L) and Plastin-3 (PLS3). EFTUD2 was found to obtain the highest number of PPI (13 PPIs), followed by AP2M1 with 4 PPIs.

Among the 18 proteins restored by H_2_S pretreatment, there is a low determined PPI. EFTUD2 and AP2M1 shown less PPIs, 8 and 3 PPIs, respectively (Figure 5A).

A number of 11 proteins were modulated by H_2_S compared to CTRL group, among which, *COP9* signalosome complex subunit 6 (*COPS6*) and *DDX5* have the highest number of PPIs (Figure 5B).

## 3. Discussion

Ever since H_2_S has been recognized as the third endogenous gaseous signaling molecule alongside CO and NO, its role in various physiological and pathological processes has been explored. In our previous study, endogenous H_2_S synthases are observed to increase in a glaucoma animal model after seven weeks of chronic elevated IOP. Furthermore, we demonstrated that administration of GYY4137, a slow-release H_2_S donor, significantly improved RGC survival under different glaucomatous injuries in vitro and in vivo [21]. The neuroprotective effect of H_2_S in the retina was partly attributed to its capability of vasorelaxation, anti-oxidative stress, neuroendocrine regulation and inflammation suppression [25] but the underlying mechanism and the specific protein interaction networks and the mechanisms involved in protective properties of H_2_S remain to be elucidated. Building on previous results, we employed a mass spectrometry-based proteomics approach to analyze the retinal proteome, and bio-informatics to algorithmically generate protein connections, which allowed us to identify the most plausible signaling pathway alterations related to H_2_S’s neuroprotective properties against ischemia-reperfusion injury in present study. This is to our knowledge, the first study to address the intricate alterations at the protein level, which can be attributed to the treatment with exogenous H_2_S, and to show how these molecules play a pivotal role in restoring retinal homeostasis against I/R injury in vivo.

Following I/R injury, the abundance of 48 proteins was significantly altered. Changes of the signaling pathways involved in mitochondrial homeostasis and function, calcium dysregulation, cytotoxicity regulation, reactive oxygen species (ROS) scavenging, neural transduction and vascular function were found. The proteins, which were regulated by H_2_S were categorized into two clusters: (1) a cluster 18 proteins, the abundance of which was significantly altered in I/R, and was then restored to normal or near normal level by H_2_S; (2) a cluster of 11 proteins, the abundance of which was significantly altered due to H_2_S administration compared to the CTRL group. Based on the results from the ingenuity pathway analysis, the signaling pathways regulated by H_2_S comprised iron homeostasis and ROS regulation, mitochondrial homeostasis and function, vasodilation and DNA repairing.

### 3.1. Changes in Iron Homeostasis and ROS Regulation

Redox mechanisms are known to partially contribute to the protective properties of H_2_S in various tissues. Firstly, we observed that the *Slc2a1* level is significantly downregulated in I/R and restored by H_2_S. *Slc2a1* facilitates docosahexaenoic acid (DHA), the oxidized form of vitamin C, transport across the inner blood-retina barrier and converts DHA to ascorbic acid and accumulates as ascorbic acid in the retina [26]. As the *Slc2a1* level is downregulated following I/R injury, vitamin-C transport is downregulated accordingly. Vitamin C is the primary circulatory antioxidant to be quickly used and depleted when excessive ROS is generated by the pathophysiological pathways triggered by I/R injury, therefore sparing other endogenous antioxidants [27]. Oxidative stress promotes the oxidation of ascorbic acid to DHA, which is shown by studies to promote neuronal death under oxidative stress [28]. Downregulated vitamin-C transport indicates that I/R injury impaired the anti-oxidative property of the retina. H_2_S increased vitamin-C transport, subsequently strengthening the anti-oxidative property of the retina as opposed to the ROS generated due to I/R.

Secondly, several proteins involved in iron homeostasis, such as biliverdin reductase A (*Blvra*), *Cyb5r3* and Glrx3, were differentially abundant due to I/R, H_2_S administration or both. Iron metabolism and regulation is crucial in mammals and is essential for physiological neuronal functions such as neural respiration and metabolic activities, myelin synthesis, neurotransmitter production and synaptic plasticity [29]. However, its overload triggers axonal degeneration and neuronal cell death [30]. Excessive iron may catalyze the formation of highly reactive hydroxyl radicals, which eventually induce the accumulation of ROS [31,32]. Nevertheless, it is well known that the disproportionally increased brain Fe level is correlated with neurodegenerative diseases such as Parkinson’s disease, Alzheimer’s disease, Huntington’s disease and neurodegeneration with brain Fe accumulation.

Most of the body’s Fe is contained within the protoporphyrin ring of heme [32]. Heme is essential for the survival of organisms, while being potentially toxic. The excessive release of heme from hemoproteins can lead to the generation of unfettered oxidative stress and programmed cell death [29]. Heme degradation is therefore important for maintaining iron homeostasis and preventing its cytotoxicity and oxidative stress [33].

The abundance of *Blvra*, the main isoform of biliverdin reductase [34], is significantly increased due to I/R injury. Increased *Blvra* level corresponds to upregulated heme degradation.

*Blvra* is a pleiotropic enzyme primarily known for reducing heme-derived biliverdin into the powerful antioxidant and anti-nitrosative molecule bilirubin [35,36]. Bilirubin is able to inhibit free radical chain reactions and protects against oxidant-induced damage [37]. Studies reported that reducing *Blvra* level in cells or knockout *Blvra* gene in mice both led to increased oxidative stress [37,38]. On the other hand, increasing the levels of *Blvra* is beneficial through an increase in intracellular bilirubin generation and direct signaling through cytoprotective pathways [34,39,40].

*Cyb5r3*, the principal reductase in mitochondria, is also significantly upregulated due to I/R injury, and can be restored by H_2_S. *Cyb5r3* is also a heme reductase and recycles oxidized heme (Fe3+) back to its reduced form (Fe2+) [41]. Upregulation of *Blvra* and *Cyb5r3* levels following I/R injury is likely to confer self-protection against oxidative stress.

Glrx3 level was the most significantly modulated by H_2_S when compared to the control group, while it was not impacted by I/R injury. Depletion of Glrx3 in mammalian cells was associated with moderate deficiencies of cytosolic Fe-S cluster enzymes and evidence of altered iron homeostasis [42]. Overexpressed Glrx3 can compensate for the lack of other reducing equivalents [43].

*Glrx3* belongs to the glutaredoxin family, it utilizes the reducing power of glutathione to maintain and regulate the cellular redox state and is essential for iron-sulfur (Fe-S) cluster assembly [44,45]. Iron-sulfur clusters are ancient, ubiquitous cofactors composed of iron and inorganic sulfur which participate in numerous biological processes [46,47,48,49].

In mammalian cells, *Glrx3* localizes to the cytosol and it has been proposed that it plays dual roles in iron trafficking and regulation [50,51]. *Glrx3* is a monothiol glutaredoxin, being able to form (2Fe-2S) cluster-bridged dimers and deliver Fe-S clusters to recipient proteins [52,53].

Detoxification of neural Fe overload is a potential therapeutic approach in the treatment of neurodegenerative disease. Based on the results of proteomic and IPA analysis, we assume that, when exposed to oxidative stress, cells increase their capability to reduce oxidized heme and to degrade it, which partially contributes to a self-protective mechanism that reduces the deleterious effects of free heme. H_2_S plays a similar reducing role as *Cyb5r3* in recycling oxidized heme to its reduced state; the abundance of *Cyb5r3* was thus not increased when H_2_S was present. Nevertheless, H_2_S has shown a strong association with *Glrx3*, which is able to regulate cellular iron homeostasis [54,55].

### 3.2. Changes in Retinal Metabolism, Mitochondrial Homeostasis and Function

As mentioned above, the abundance of *Slc2a1* is significantly downregulated in I/R and restored by H_2_S to near normal levels, which also corresponds to increased HIF1α signaling. *Slc2a1* abundance in HIF1α signaling plays a key role in promoting neuronal survival by mediating the endogenous protective responses after hypoxia-ischemia. In glaucoma mice, decreased HIF1α expression is observed to be correlated with RGC loss [56]. H_2_S regulates HIF1 factors in different patterns, depending on different cell types and experimental conditions. It increases expression of the HIF1α in rat brain capillary endothelial cells and in mouse spinal-cord primary culture [57,58], and inhibits HIF1 activation in human hepatoma Hep3B cells, cervical carcinoma HeLa cells and aortic smooth-muscle cells [59,60].

Our proteomic data indicate that H_2_S administration increased HIF1α signaling. Upregulating expression of HIF1α induces an increase of aerobic glycolysis, which transforms glucose to lactate and generates nicotinamide adenine dinucleotide (NAD+) [61]. HIF1 also directly induces pyruvate dehydrogenase kinase 1 [62], which phosphorylates the pyruvate dehydrogenase (PDH) complex, and consequently inhibits PDH complex from catalyzing pyruvate to form acetyl CoA. This fuels the mitochondrial tricarboxylic acid cycle (TCA) cycle, which provides the mitochondrial NADH needed to power electron transport. Hence, HIF1α actively represses mitochondrial respiration.

Furthermore, in this study, H_2_S reduced the abundance of NADH dehydrogenase (ubiquinone) 1 alpha subcomplex subunit 5 (*Ndufa5*), which subsequently also decreased mitochondrial respiration. *Ndufa5*, localizes to the inner mitochondrial membrane and is an accessory sub-unit of complex I [63]. Reducing *Ndufa5* suppresses complex I activity [64,65]. It is generally believed, that mitochondrial complex I deficiency contributes to the process of neurodegeneration because of reduced adenosine triphosphate (ATP) production, oxygen consumption and increased ROS production. However, complex I has been reported as a site of mitochondrial ROS generation [66,67], and actively suppressing its activity may actually be protective. Neuronal complex I deficiency is not correlated with the neurodegeneration, mitochondrial DNA damage in Parkinson’s disease brain, nor in mice with a central nervous system knockout of *Ndufa5* [64,68]. It has been shown in neurodegenerative disorders that reversible inhibition of complex I can be protective against ischemia-reperfusion injury by modifying mitochondrial ROS production [69,70].

Studies strongly suggest that active suppression of the TCA cycle, mitochondrial respiration and ROS production, as well as augmentation of ATP levels, are crucial for the cell survival under I/R injury [62,71]. We assume that by suppressing mitochondrial respiration through upregulating HIF1-α and downregulating *Ndufa5* levels, H_2_S reduces mitochondrial oxidative phosphorylation, thereby repressing mitochondrial oxygen consumption and resulting in the relative increase of intracellular oxygen tension and limited ROS production under I/R.

As discussed above, mitochondrial PDH activity is attenuated by HIF1α following H_2_S administration. Neurons usually prefer glucose for energy, but when mitochondrial metabolism of pyruvate is limited, ketone bodies can be oxidized and release the only alternative source of acetyl CoA for neurons [72]. Using ketone bodies as an alternative energy source has shown neuroprotective effects in Alzheimer’s disease, Parkinson’s disease and ischemic and traumatic brain injury [73,74,75,76,77,78]. The metabolic pathway that produces ketone bodies is ketogenesis. In this study, the signaling pathways most significantly downregulated by I/R injury and then restored by H_2_S were ketogenesis, ketolysis and leucine degradation.

In comparison to glucose, ketone bodies have a higher inherent energy [79]. As ketone bodies provide more energy per unit of oxygen than glucose [80,81], in states of metabolic stress such as I/R injury, ketone bodies are more energy-efficient fuel for neurons. They can also bypass the inhibited PDH complex and maintain the metabolites of TCA cycle, therefore permitted continued ATP production under ischemia.

Administration of H_2_S has shown evidence of actively suppressing oxidative phosphorylation and limiting the utilization of glucose as the energy source, therefore increasing intracellular oxygen tension under ischemia and limiting ROS production during reperfusion. Furthermore, H_2_S promotes the utilization of ketone bodies as an alternative energy source, which is more energy-efficient than glucose and maintains ATP production. H_2_S therefore enhanced the ability of retinal neurons to withstand metabolic stress induced by I/R, which would normally deplete the resilience of the neurons and result in neuronal cell loss (see in Figure 6).

### 3.3. Changes in Retinal Vascular Function

Decreased average blood flow in the retina, optic nerve head and choroidal circulations is also demonstrated in glaucoma patients [82,83,84]. The “vascular theory” of glaucoma hypothesizes RGC loss as a consequence of insufficient blood supply [85]. Vasospasm and autoregulatory dysfunction have been postulated to reduce ocular blood flow [86]. Increased IOP has also been shown to reduce vascular caliber in rat retina while H_2_S acted as a vasodilator in our previous studies [21]. H_2_S can be endogenously generated in vascular smooth muscle cells. Its vasorelaxant potency is partially mediated by a functional endothelium [87].

However, I/R is known to cause endothelial dysfunction. In this study, eNOS signaling is downregulated following I/R injury. Endothelium-derived NO is a critical regulator of vascular homeostasis and tone [88]. NO continually regulates the diameter of blood vessels and maintains an anti-apoptotic environment in the vessel wall [89].

Downregulation of eNOS signaling is an indicator of impaired endothelial function, which results in diminished microcirculation and reduced response to endothelial-dependent vasodilators and vasoconstrictors. This causes dysregulated blood flow and loss of the endothelial barrier.

While eNOS signaling is downregulated, protein kinase A (PKA) signaling is significantly upregulated following I/R. PKA represents a signaling hub for a large variety of hormones, neurotransmitters and cytokines [90]. PKA has been shown to regulate different aspects of endothelial cell physiology [91,92,93], and to inhibit angiogenesis when activated [94,95]. Upregulation of PKA is likely a self-protective mechanism.

Furthermore, one of the most significantly upregulated proteins I/R, which can be restored by H_2_S to normal levels, is *Cyb5r3*. It is the principal reductase involved in the mitochondrial amidoxime reducing component (mARC)-containing enzyme system [96]. In vascular smooth muscle, *Cyb5r3* functions as a soluble guanylyl cyclase (sGC) heme iron reductase, and is critical for vasodilation [97]. Through its reductase activity, *Cyb5r3* maintains NO-sGC-cGMP function [98]. The NO-sGC-cGMP pathway in aortic smooth muscle cell (SMC) is known to relax SMC and dilate vessels [98].

Hydrogen sulfide and the −SH anion reduce a variety of organic substrates [99]. It is reasonable to assume that H_2_S plays a similar reducing role as *Cyb5r3* in maintaining vascular relaxation and enabling a less constricted vascular environment due to I/R.

Although the abundance of *DDX5* was not influenced by I/R injury, it is significantly more abundant following the administration of H_2_S. As *DDX5* regulates the expression of a number of key regulators upstream of the estrogen-receptor [100], estrogen-receptor signaling was also significantly upregulated in our PPI results.

Increasing evidence suggests that estrogen exposure may have a neuroprotective effect on the progression of glaucoma and may alter its pathogenesis [101]. The vasodilation caused by estrogen and its effects on aqueous humor outflow may contribute [102]. These estrogen receptors are abundantly expressed throughout the eye, and in particular the retina and more specifically in the RGC [103,104]. In various CNS injuries and diseases, the activation of estrogen receptor signaling has shown neuroprotective effects via attenuation of neuro-inflammation and neurodegeneration [105]. In different animal models of RGC degeneration, activation of estrogen receptor attenuated RGC apoptosis under acute elevated IOP [106]; impaired estrogen receptor expression is observed in a mouse model with early RGC apoptosis [107]; and in Leber’s hereditary optic neuropathy cells activating estrogen receptors improve cell viability by reducing apoptosis [108]. Moreover, inhibition of estrogen synthesis is associated with IOP elevation and reduced RGC counts in female mice [109] and blockage of estrogen receptor signaling in rats intensifies impairment in visual function and retinal structure after ocular hypertension [110]. Although estrogen receptor signaling was not impacted by I/R injury in our study, activating estrogen receptor signaling via upregulating *DDX5* is still one of the possible pathways through which H_2_S exerts protective effects and partially contributes to its capability of vasorelaxation, anti-oxidative stress and inflammation suppression.

H_2_S also significantly downregulated annexin A6 compared to the control group, which is a calcium dependent phospholipid binding protein [111].

Annexins are important in Ca(2+)-induced neurotoxicity or neuroprotection [112]. Annexin A6 has been shown to play a central role in vascular remodeling processes [113]. It has been demonstrated that annexin A6 accumulates and converts exosomes in vascular smooth muscle cells into calcifications under calcium stress, which is known to be induced by I/R [113]. Vascular smooth muscle cells contribute significantly to physiological regulation of vascular tone and arterial blood pressure [114]. The presence of coronary artery calcification is significantly associated with raised IOP regardless of conventional cardiovascular risk factors [115]. By reducing the abundance of annexin A6, H_2_S potentially makes the calcification in vascular smooth muscle cells subside due to I/R, thus maintaining vascular autoregulation.

PKA signaling and *Cyb5r3* level is upregulated, likely serving as a self-protective mechanism to maintain endothelial function and vasodilation against I/R injury, while eNOS signaling is downregulated as an indication of endothelial dysfunction. Administration of H_2_S regulated the abundance of *Cyb5r3* and activated estrogen receptor signaling to facilitate vascular relaxation. H_2_S also reduced the abundance of annexin A6, which plays a central role in artery calcification. Combined together, H_2_S enabled a less constricted vascular environment in the retina, thereby resulting in better retinal perfusion to counteract I/R injury (see Figure 7).

### 3.4. Changes in GABA Receptor Signaling

GABA receptor signaling is the most impacted signaling pathway by I/R and this can be restored by H_2_S. The *AP2M1* and *SLC32A1* levels are upregulated due to I/R, which correspond to upregulated GABA receptor signaling. gamma-aminobutyric acid (GABA) is the main inhibitory neurotransmitter in the mammalian brain [116]. In the retina, GABA is similarly important in neural inhibition with approximately 40% of all retinal cells having been described as GABAergic [117]. GABA-containing vesicles release the neurotransmitter into the synaptic cleft to bind with GABA receptors present on the post-synaptic neuron [118].

*SLC32A1*, solute carrier family 32 (GABA vesicular transporter) member 1, is a transmembrane transporter which is responsible for the storage of GABA or glycine in synaptic vesicles [119]. *AP2M1* is highly expressed in the central nervous system [120,121], and encodes the μ-subunit of the adaptor protein complex 2 (*AP-2*) [122]. AP-2 regulates the neuronal surface levels of GABA and glutamate receptors [123,124]. An imbalance between excitatory and inhibitory synaptic transmission is thought to contribute to excitotoxicity and neuronal cell death during ischemic insult. AP-2 was found to be crucial in reducing the loss of synaptic GABA receptors during simulated ischemia in the rat brain [125]. Increased abundance of *SLC32A1* and *AP2M1* suggests raised retinal GABA levels following I/R injury, which may occur in order to maintain excitatory/inhibitory balance, thereby improving the survival of RGCs, the latter being most susceptible to extracellular glutamate.

However, similar to glutamate, the functions of GABA are also controversial. There is evidence that the retina is extremely sensitive to abnormally accumulated GABA and its resulting toxicity, particularly in the presence of ischemia [126,127,128]. An antiepileptic drug, vigabatrin (gamma-vinyl GABA), exerts its effect by increasing GABA levels [129]; as the drug increases retinal GABA levels much more significantly than in the brain and retina has a lower tolerance to GABA toxicity than the brain [127,128], one third of patients receiving vigabatrin present binasal visual field loss [130].

GABA also has a role in the regulation of vascular tone [127]. GABA receptors are known to interact with perivascular astrocytes which synthesize vasoactive materials [131,132]. In the rat retina, GABA is shown to be capable of eliciting both vasodilator and vasoconstrictor responses but endogenous GABA is unlikely to be an important regulator of resting vascular diameter and blood flow in the retina [133].

The exact role of GABA receptor signaling in this study is unclear but the results certainly raise intriguing questions about the involvement of GABA receptor signaling in neurodegeneration and the interaction between GABA signaling and H_2_S.

### 3.5. Changes in DNA Repair

The nucleotide excision repair (NER) pathway is at the top of the canonical pathways modulated by H_2_S, with a particularly significant upregulation of *COPS6* level. In combination with the ubiquitin (Ub) proteasome system (UPS) the COP9 signalosome controls the stability of cellular regulators [134] and regulates several important intracellular pathways, including DNA repair, cell cycle, developmental changes, and some aspects of immune responses [135,136,137].

*COPS6* has been shown to be cleaved by caspase 3 during apoptosis in vitro and in vivo [138] and this cleavage can be completely blocked by specific caspase 8 inhibitor [139]. Upregulation of *COPS6* led to upregulation of the NER pathway, one of the major cellular DNA repair pathways for the removal of bulky helix lesions [140,141]. Enhancement of the NER pathway has shown protective effects in peripheral neurons both in vitro and in vivo [142]. The roles of DNA repair in neuronal cell survival and the response to aging and ROS such as that generated by mitochondrial respiration in glaucoma is of particular interest. The promotion of DNA repair by activation of the NER pathway through COP9 signalosome subunit upregulation could be an interesting aspect to explore in future studies.

Neurodegenerative diseases are characterized by decades of apparent normality, during which local deficits are compensated. Eventually the deficits become too accentuated or the compensatory mechanisms fail [143]. From the results we obtained in this study, I/R injury led to changes in ROS regulation, retinal metabolism, mitochondrial homeostasis and function, retinal vascular function and metal homeostasis, with over half of the altered pathways in the I/R group being self-protective mechanisms. H_2_S provided additional support where self-protective mechanisms fell short and demonstrated DNA repair properties, eventually providing neurons a better outcome against I/R injury.

Admittedly there are limitations in this study. Firstly, individual retinal samples were deliberately pooled into biological replicates to minimize inter-individual variations [144,145]. The exact role H_2_S plays in neurodegeneration within the retina is largely vague and still awaits to be thoroughly elucidated. Due to the limitation of sample material and the variety of the signaling pathways H_2_S is involved in, we cannot confirm the differentially expressed proteins by a second technique in the present study design. As the foremost study providing a thorough overview of the retina proteome changes related to neuroprotective properties of H_2_S in the retina, the main focus was to acquire a comprehensive perspective of the complex interaction between different proteins. Building on these results, different techniques will definitely be used to confirm the differentially expressed proteins and altered signaling pathways in our future studies.

Secondly, the results attained from the ischemia/reperfusion model are not necessarily valid in other glaucomatous models. There are various experimental models of glaucoma in the marketplace; each of them has specific pros and cons. It seems that degeneration caused by chronic elevated IOP mimics the characteristics of glaucoma better than I/R model. Clinical observations have demonstrated retinal vascular abnormalities and impaired blood flow at the optic nerve head which suggest ischemia plays a key role in the pathogenesis of glaucoma [146,147]. A pure ischemic lesion certainly cannot represent the glaucomatous damage; reperfusion injury is also present in glaucoma patients [148]. It is evident that IOP fluctuations are more damaging than a stabled increased IOP, and reduced circulation due to vascular dysregulation is more damaging than reduced circulation due to arteriosclerosis [149].

This study provides, for the first time, an overall insight into the mainstay retinal proteins and pivotal signaling pathways that interact with H_2_S to maintain retinal homeostasis against I/R injury. In particular, it reveals that exogenous H_2_S activates the pathways linked to iron regulation, ROS scavenging, the modulation of mitochondrial homeostasis and function, maintenance of retinal vascular function and GABA receptor signaling. H_2_S may be a promising and effective therapeutic tool for glaucoma. The findings presented here point to directions for further research, for example, exploring the significance of these identified proteins and pathways and investigating related data from other glaucoma models.

## 4. Materials and Methods

The overview of sampling and proteomics workflow used in this study is presented in Figure 8.

### 4.1. Animals

Female Sprague–Dawley rats (*n* = 12; 250–300 g) at the same age were used for glaucoma animal model in this study. The use of animals for research purposes was approved by the Health Investigation Office Rhineland-Palatinate (permission number: 14-1-085; approvals date: 13 October 2014). All experimental procedures were in accordance with the Association of Research in Vision and Ophthalmology (ARVO)—Statement for the Use of Animals in Ophthalmic and Vision Research, and the guidelines of the Institutional Animal Care and Use Committee. Animals were kept under a day- and night-cycle of 12 h at the Translational Animal Research Center (TARC) of the University Medical Center of Johannes-Gutenberg University Mainz.

Minimizing the discomfort and pain for the animals was prioritized in the experimental interventions. A mixture of medetomidine hydrochlorid (Dorbene vet., Pfizer, New York, NY, USA) and ketamine (Inresa Arzneimittel, Freiburg, Germany) was administered intraperitoneally for anesthesia, and oxybuprocain (Novesine, OmniVision, Puchheim, Germany) was applied topically to ocular surfaces. To reduce post operation pain, novaminsulfon (Novalgin, Ratiopharm, Ulm, Germany) was injected subcutaneously after intervention. Health condition and general behavior of the animals were observed daily by TARC staff.

### 4.2. Intravitreal Injection of GYY4137

Animals were anesthetized as described above, following topical anesthesia, 3 μL of GYY4137, a H_2_S slow-releasing donor (SIGMA-ALDRICH; Darmstadt, Germany) was injected into the vitreous body with Hamilton syringe (SIGMA-ALDRICH; Darmstadt, Germany) paired with 33-gauge needle. The injection volume was 3 μL to achieve optimal distribution of the compound and minimum disruption to the IOP [150]. The injection needle was retained intravitreally for 15 s to avoid reflux. The intraocular concentration of GYY4137 was approximately 100 nM as average vitreous volume of an adult rat eye is approximately 56 uL [150].

### 4.3. Ischemia-Reperfusion Injury Model

The retinal ischemia-reperfusion (I/R) injury model is a well-established animal model which imitates the clinical manifestations of retinal vascular occlusion diseases and acute glaucoma. It has also been widely adapted to study retinal neuronal cell damage. Female Sprague-Dawley rats (*n* = 12; 250–300 g) were anesthetized, with six receiving an intravitreal injection of GYY4137 in the left eye and the other six receiving intravitreal injections of saline, as described as above. The saline injection group served as the corresponding control with retinae from untouched contralateral eyes being assigned as baseline controls. Following the intravitreal injection, the anterior chamber was entered from the superotemporal cornea with a 30-gauge needle (see Figure 8A). The needle was connected to a sterile saline container, which was lifted until IOP was elevated to 55 mmHg for a period of 60 min. IOP was measured by a rebound-Tonolab (iCare, Vantaa, Finland) for rodent. Central cornea, lens and other surrounding tissues were carefully avoided and not injured in the process. The iris turned pale and the retina lost its red reflex during the procedure, thus confirming the ischemic condition of the eye. Animals were kept alive for 24 h after the intervention for sufficient reperfusion injury.

### 4.4. Preparation of Retinal Explants

Sprague-Dawley rats were sacrificed under CO_2_ atmosphere. Eyes were explanted immediately post-mortem and retinae were harvested and flat-mounted as previously described [21]. In short, the eye is dissected into the anterior segment and the optic cup. Intact retina was harvested from the optic cup and flat-mounted on the Millipore filters (Millipore; Millicell, Cork, Ireland) with the ganglion cell side up, vitreous body was then carefully removed (Figure 8B).

### 4.5. Quantification of Retinal Ganglion Cells

One quarter of each retinal explant was carefully separated under microscope for subsequent immunohistochemical staining against the brain-specific homeobox/POU domain protein 3A (Brn3a). Unlike Thy1, which is another RGC-specific antigen, the expression pattern of Brn3a does not change after retinal injury and the levels of Brn3a protein are decreased with time after retinal injury, which is in agreement with the loss of RGC [151]. Therefore, Brn3a immunodetection is a powerful tool to assess RGC survival in rat injury models [152,153]. Retinal tissue (*n* = 6 quarters/group) was fixed and stained as previously described [154]. Immunofluorescent RGCs were visualized and photographed with a fluorescent microscope (Carl Zeiss, Ltd., Hertfordshire, UK). Images of retinal whole mounts were photographed from 4 different regions of each quadrant of the retina (Figure 1) using a Zeiss fluorescent microscope (Carl Zeiss, Ltd., Hertfordshire, UK). Images were captured at a 20-fold magnification using a fluorescent camera (Carl Zeiss, Ltd., Hertfordshire, UK). Total numbers of Brn3a-positive cells were counted with the assistance of ImageJ (ImageJ Fiji v_1, total = 4 counts/quadrant, and six retinal quadrants per treatment group). The mean number of RGCs per quadrant was calculated from a mean count at each of the 4 different regions.

### 4.6. Mass Spectrometry Sample Preparation

The remaining retinal explants were rinsed in ice-cold PBS to remove contaminants and weighed immediately thereafter. Retinal protein extraction from each piece of retina was carried out using T-PER tissue protein extraction reagent (Thermo Scientific Inc., Waltham, MA, USA) following an in-house established method, as described by Manicam et al. [155]. Protein concentration of each eluate was measured by standard bicinchoninic acid (BCA) protein assay kit (Pierce, Rockford, IL, USA) as per manufacturer’s instructions. Following protein measurements, retinal protein samples from respective groups were pooled equally into three biological replicates, represented by R1, R2 and R3. From each replicate, 50 μg protein was subjected to polyacrylamide gel electrophoresis (PAGE). Protein lanes were sliced into 25 bands per replicate and destained, reduced, alkylated, dehydrated and subjected to in-gel trypsin digestion at 37 °C overnight. The peptides were then extracted firstly with acetonitrile and then with a mixture of 5% formic acid and acetonitrile 1:2 (*v*/*v*), the supernatant was pooled. The extracted peptides were further purified by SOLAμ SPE plates and cartridges (Thermo Scientific Inc., Waltham, MA, USA) following manufacture’s instructions. The eluate was dried in SpeedVac and dissolved in 10 μL of 0.1% trifluoroacetic acid (TFA) for LC-MS/MS analysis.

The continuum MS data was collected by an ESI-LTQ Orbitrap XL-MS system (Thermo Scientific, Bremen, Germany). The tandem MS spectra were searched against UniProt database (*Homo sapiens*; *Rattus*; 21.11.2018). A target-decoy-based false discovery rate (FDR) was set to 0.01 for identification of peptide and protein.

(Figure 8C–F. The detailed sample preparation procedure and general parameters of the instrument can be found in Appendix A “Mass Spectrometry Sample Preparation” and Appendix A “MaxQuant parameters”)

### 4.7. Bioinformatics, Functional Annotation and Pathways Analyses and Statistics

Data generated from MaxQuant analysis was subjected to further statistical analysis, carried out with Perseus [156]. Briefly, the data were filtered with a minimum of three valid values in at least one group. The missing values were replaced from normal distribution (width, 0.3; down shift, 1.8) in Perseus. All label-free quantification (LFQ) intensities were subject to a log2 transformation. Followed by a Student’s two-sample t-test for the group’s comparison with *p* < 0.05 to identify the significantly differentially expressed proteins. Unsupervised hierarchical clustering analysis was done with the z-scores of the LFQ intensities following Euclidean distance (linkage = average; preprocess with k-means) and a heat map is generated to demonstrate the result.

Furthermore, the gene names of significantly differentially expressed proteins (*p* < 0.05) in each group were subject to functional annotation and pathways analyses by Ingenuity Pathway Analysis (v01-04, IPA; Ingenuity QIAGEN, Redwood City, CA, USA) (https://www.qiagenbioinformatics.com/products/ingenuity-pathway-analysis). Using Benjamini–Hochberg multiple testing correction (−log B–H > 1.3), top canonical pathways of the differentially expressed proteins were presented with *p*-values calculated (Figure 8G,H).

Prism 8 software (GraphPad Software, Inc., San Diego, CA, USA) was used in this study for statistical calculations of RGC quantification and displaying of the figures. The averaged RGC density was calculated per mm^2^. Data values are expressed as the mean ± SD. Significance of difference between groups was determined by 1-way ANOVA. Results were considered to be statistically significant when *p* < 0.05 (Figure 8I).

## Figures and Tables

**Figure 1 pharmaceuticals-13-00213-f001:**
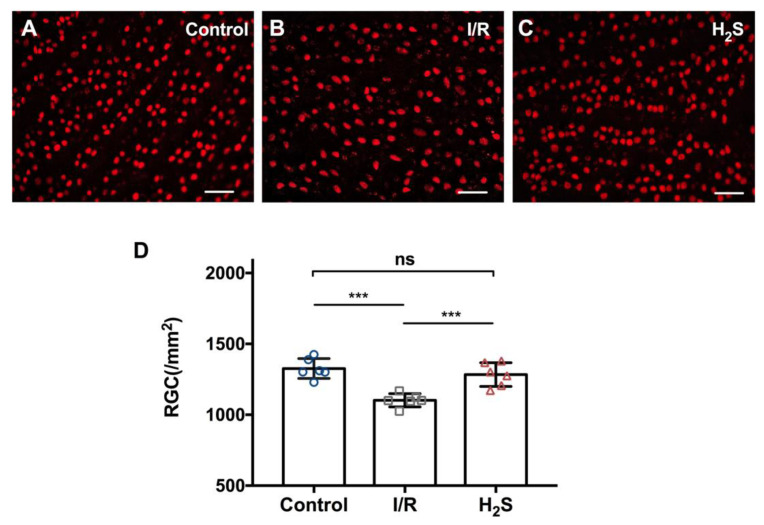
Pre-treatment with H_2_S protects RGC against ischemia-reperfusion injury in vivo. (**A**–**C**) Representative fluorescence microscopy of Brn3a staining of retinal explants 24 h after inducing ischemia/reperfusion injury in vivo. (**D**) Compared to the contralateral control (1310.6 ± 236.4 RGC/mm^2^), ischemia-reperfusion injury resulted in significant RGC loss in experimental eyes (1101.7 ± 116.4 RGC/mm^2^). Pretreatment with GYY4137 showed a significant reduction of RGC loss in vivo (1295.4 ± 136.1 RGC/mm^2^). Total number of RGC between control and H_2_S treated group is not significantly different (*** *p* < 0.0005, *n* = 6/group, means ± SD, scale bar = 50 μm).

**Figure 2 pharmaceuticals-13-00213-f002:**
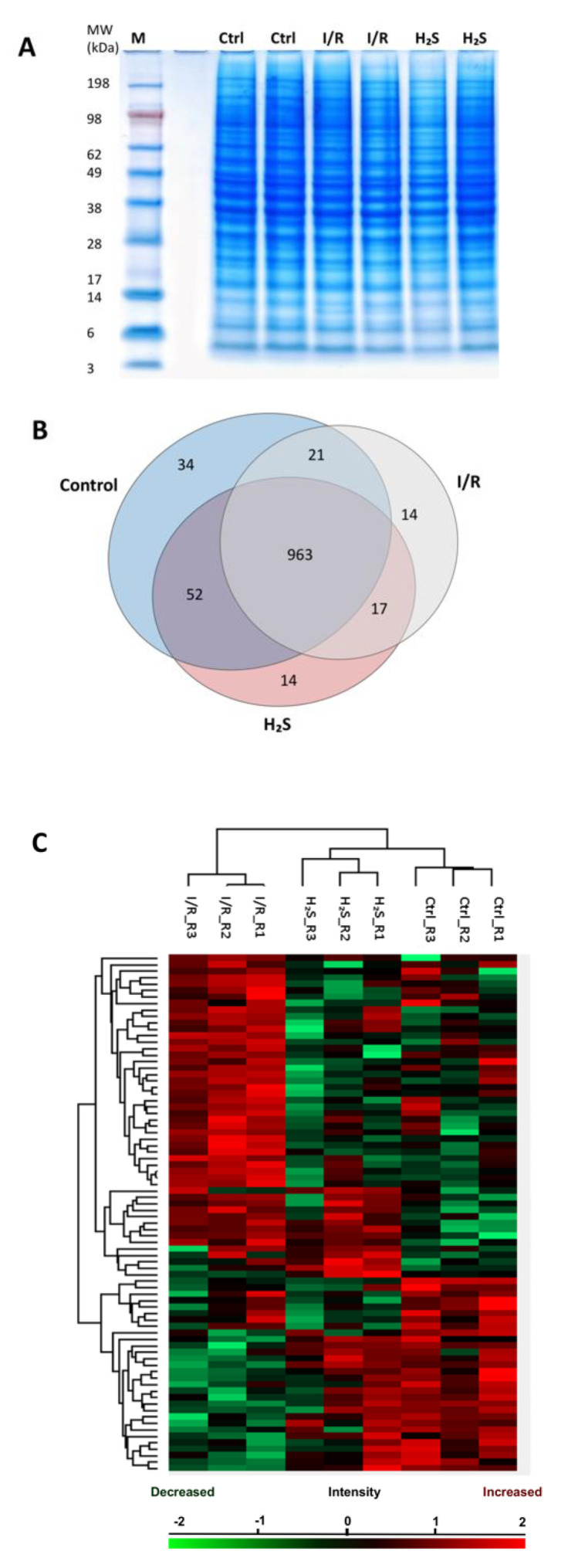
Retinal proteome of Ischemia-reperfusion (I/R) injured and H_2_S pre-treated retinae. (**A**) Representative retinal protein profiles of both I/R injured and H_2_S pre-treated retinae compared to non-operated retinae (designated as CTRL) resolved in PAGE gel stained with colloidal blue. M: marker. (**B**) Venn diagram depicting overlaps of identified retinal proteins in I/R injured and H_2_S pre-treated groups compared to the control group. (**C**) Heat map depicts the hierarchical clustering of the differentially expressed retinal proteins in I/R injured and H_2_S pre-treated group compared to the control group, H_2_S pre-treat group demonstrated a more similar protein makeup to the control group (detailed list of proteins can also be found in Appendix A).

**Figure 3 pharmaceuticals-13-00213-f003:**
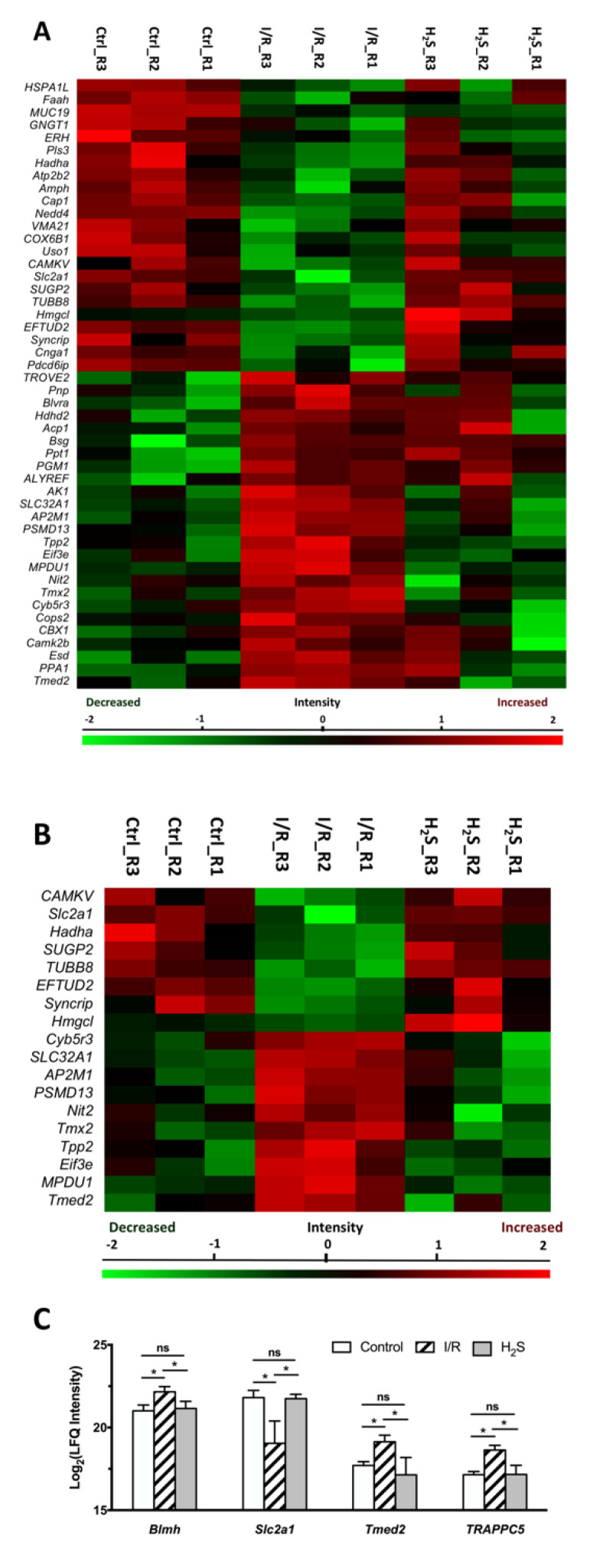
Differential expression profiles of retinal proteins in control, I/R and H_2_S. The relative protein abundance is represented as heat maps of each protein clusters. (**A**) A cluster of 48 proteins, the abundance of which were significantly altered due to I/R injury (also see Table 1); (**B**) a cluster of 18 proteins, the abundance of which were significantly altered due to I/R injury and restored by H_2_S (Table 2); (**D**) a cluster of 11 proteins, which were significantly differentially expressed in H_2_S group (Table 3). Charts showing the different expression profiles of (**C**) some of the significantly (* *p* < 0.05) dysregulated proteins in I/R group that were restored due to administration of H_2_S, (**E**) some of the significantly differentially expressed retinal proteins in H_2_S group.

**Figure 4 pharmaceuticals-13-00213-f004:**
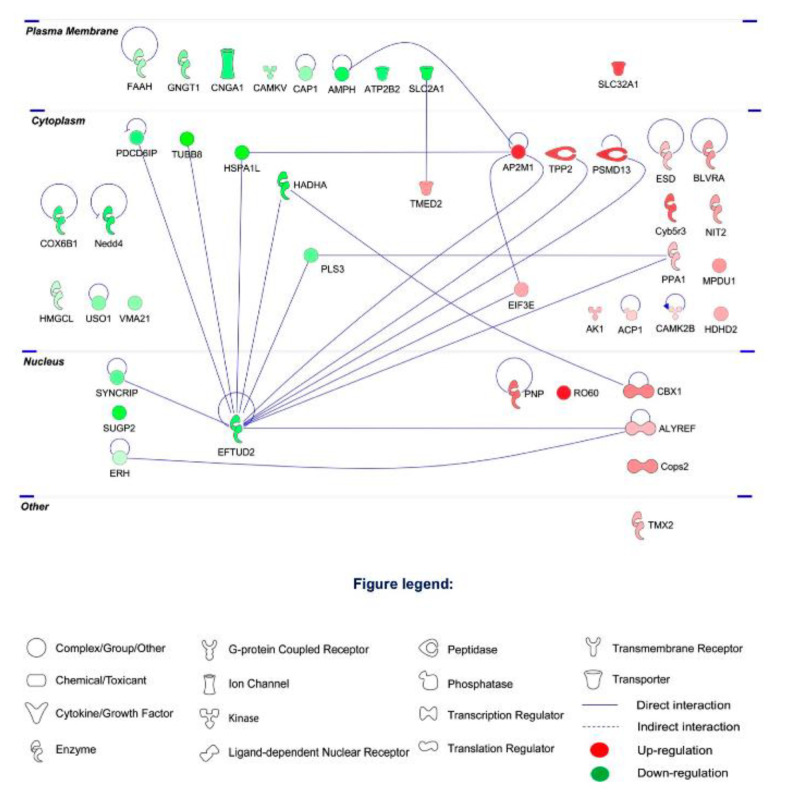
Protein–protein interaction networks of the differentially expressed retinal proteins due to I/R comparing to CTRL group. The major interaction networks of differentially expressed retinal proteins obtained by IPA analysis (QIAGEN Inc., https://www.qiagenbioinformatics.com/products/ingenuity-pathway-analysis) in I/R group.

**Figure 5 pharmaceuticals-13-00213-f005:**
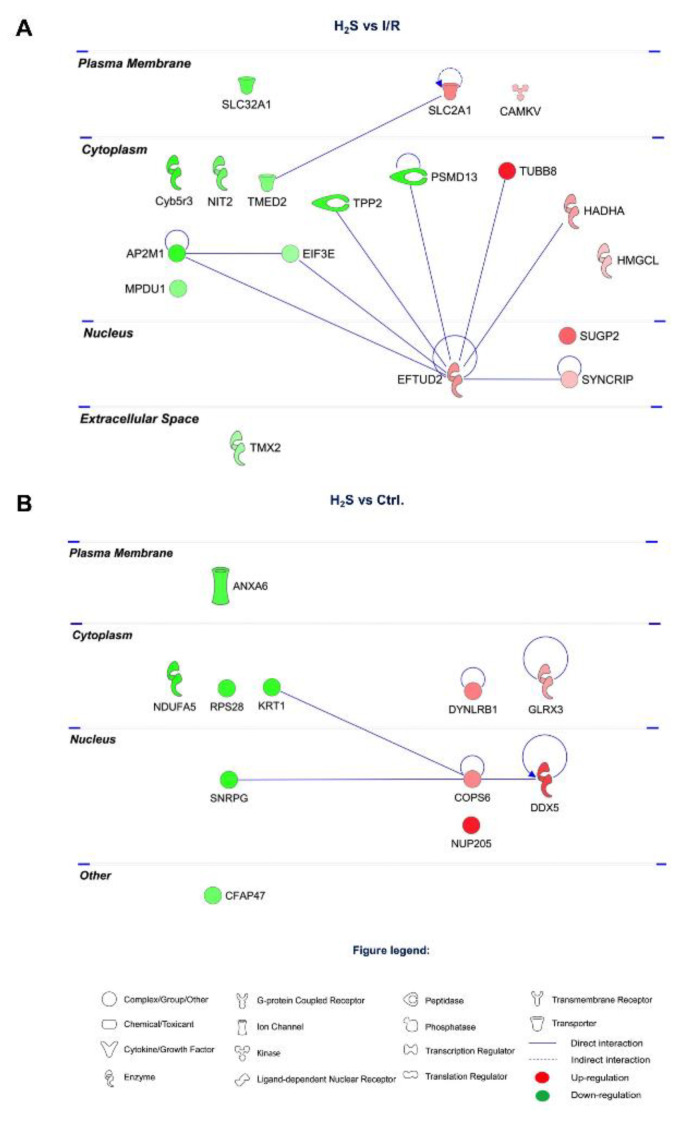
Protein–protein interaction networks of the differentially expressed retinal proteins modulated by H_2_S. (**A**) 18 proteins restored by H_2_S comparing to I/R group. (**B**) 11 differentially expressed retinal proteins modulated by H_2_S comparing to CTRL group. The major interaction networks of restored retinal proteins obtained by IPA analysis in H_2_S group.

**Figure 6 pharmaceuticals-13-00213-f006:**
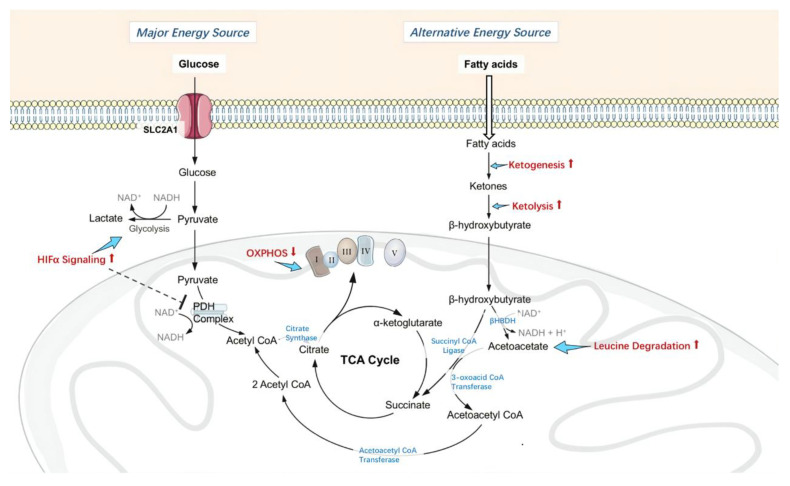
Changes in retinal metabolism, mitochondrial homeostasis and function. Administration of H_2_S actively inhibited pyruvate dehydrogenase (PDH) complex activity by upregulating HIF1α signaling, therefore limited using glucose as energy source, and suppressed oxidative phosphorylation by inhibiting Complex I activity, consequently increased intracellular oxygen tension under ischemia and limited ROS production during reperfusion. Furthermore, H_2_S promotes the utilization of ketone bodies as an alternative energy source, which is more energy-efficient than glucose, thus permitted continuous adenosine triphosphate (ATP) production. H_2_S enhanced the ability of retinal neurons to withstand metabolic stress induced by I/R, therefore less neuronal cell loss.

**Figure 7 pharmaceuticals-13-00213-f007:**
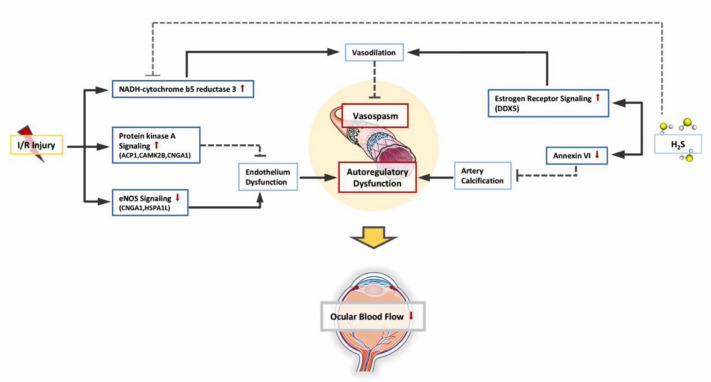
Changes in retinal vascular function. Due to I/R, PKA (protein kinase A) signaling and NADH-cytochrome b5 reductase 3(*Cyb5r3*) is upregulated as self-protective mechanism to maintain endothelial function and vasodilation, while eNOS signaling is downregulated as an indication of endothelial dysfunction. Administration of H_2_S activated estrogen receptor signaling to facilitate vascular relaxation. H_2_S also reduced the abundance of annexin A6, which plays a central role in artery calcification. Combined together, H_2_S protected the blood flow regulatory mechanisms and enabled a less constricted vascular environment in retina against I/R injury.

**Figure 8 pharmaceuticals-13-00213-f008:**
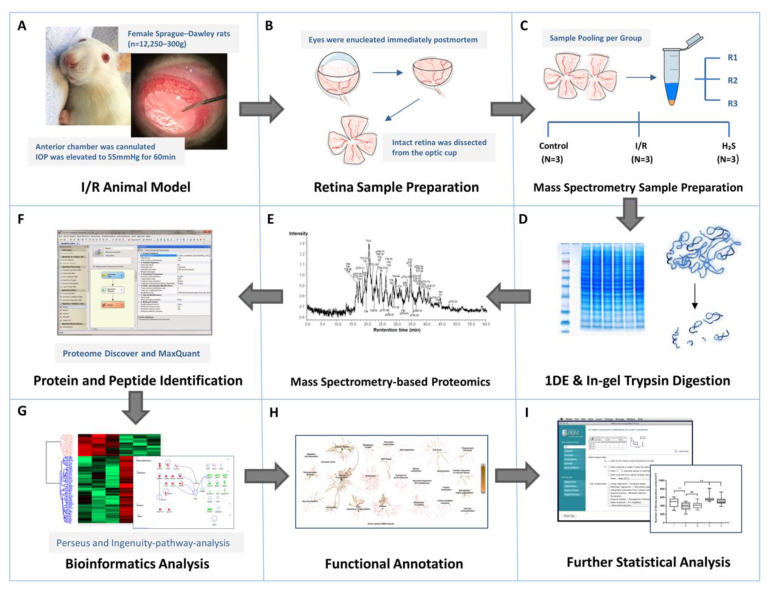
Workflow overview. (**A**) I/R injury was induced in the left eyes of adult female Sprague-Dawley rats (*n* = 12), six of which received intravitreal injection of GYY4137, an H_2_S slow-releasing precursor, shortly before intervention. (**B**) Animals were executed 24 h after intervention, retinae were harvested immediately postmortem. (**C**–**E**) Retinae from contralateral eyes were designated as controls. Retinal samples were immediately weighed and lysed by T-PER tissue protein extraction reagent and Bullet Blender Storm. Six retinal protein samples from respective groups were pooled equally into three biological replicates after protein measurements, represented by RI, R2 and R3 (*N* = 3 replicates per group), and subsequently subjected to PAGE. The protein bands were sliced and digested by trypsin prior to proteomic analysis by LC-ESI-MS/MS. (**F**–**H**) The emerging datasets were subjected to robust bioinformatics analyses and functional annotations to identify the differential protein expressions and protein interaction networks. (**I**) To confirm the result of I/R injury and protective effect of H_2_S in retina, one quarter of each retinal sample (*n* = 6 per group) were subjected to immunohistochemistry staining against Brn3a for RGC quantification. The averaged RGC density was calculated per mm^2^. Significance of difference between groups was determined by 1-way ANOVA.

**Table 1 pharmaceuticals-13-00213-t001:** List of the 48 significantly differentially expressed retinal proteins identified in I/R group.

Protein IDs	Gene Names	*p*-Value	Abundance
Q15029	*EFTUD2*	0.0006042	low
Q62940	*Nedd4*	0.00061229	low
Q08163	*Cap1*	0.00086462	low
Q7Z5P9	*MUC19*	0.00091244	low
Q9H598	*SLC32A1*	0.00091751	high
Q15181	*PPA1*	0.0016681	high
Q3ZCM7	*TUBB8*	0.0031885	low
Q96CW1	*AP2M1*	0.00352499	high
O75352	*MPDU1*	0.00377415	high
P97519	*Hmgcl*	0.00470492	low
P34931	*HSPA1L*	0.00595437	low
P20070	*Cyb5r3*	0.00814946	high
P36871	*PGM1*	0.00912329	high
B0BNE5	*Esd*	0.0096494	high
Q9UNM6	*PSMD13*	0.0112771	high
Q5XIK2	*Tmx2*	0.0124046	high
P46844	*BLVRA*	0.0135271	high
Q63524	*Tmed2*	0.0150562	high
Q63598	*Pls3*	0.0158343	low
Q62927	*Cnga1*	0.0175332	low
Q8IX01	*SUGP2*	0.019042	low
P97612	*Faah*	0.0195873	low
P61203	*Cops2*	0.0208293	high
Q497B0	*Nit2*	0.0211131	high
P39069	*Ak1*	0.0216897	high
P11506	*Atp2b2*	0.0227347	low
Q7TP47	*Syncrip*	0.0239965	low
Q8NCB2	*CAMKV*	0.0247766	low
P14854	*COX6B1*	0.0270203	low
P11167	*Slc2a1*	0.0276784	low
P45479	*Ppt1*	0.0279009	high
Q9QZA2	*Pdcd6ip*	0.0299058	low
P41498	*Acp1*	0.032686	high
P10155	*TROVE2*	0.0333496	high
Q64428	*Hadha*	0.0337918	low
P83916	*CBX1*	0.0342579	high
Q64560	*Tpp2*	0.0352463	high
P26453	*Bsg*	0.0366156	high
Q641X8	*Eif3e*	0.0387402	high
Q6AYR6	*Hdhd2*	0.0399141	high
P08413	*Camk2b*	0.0416695	high
O08838	*Amph*	0.0425382	low
Q86V81	*ALYREF*	0.0427918	high
P63211	*GNGT1*	0.043069	low
P84090	*ERH*	0.0437723	low
P41542	*Uso1*	0.0456321	low
Q3ZAQ7	*VMA21*	0.0494713	low
P85973	*Pnp*	0.0496038	high

**Table 2 pharmaceuticals-13-00213-t002:** List of 18 retinal proteins which were restored in H_2_S treated group. Proteins were classified into different categories by PANTHER classification system.

Protein Class	Gene Name	*p*-Value	Abundance
Cytoskeletal Protein	*TUBB8*	0.0031885	low
Membrane Traffic Protein	*AP2M1*	0.00352499	high
	*Tmed2*	0.0150562	high
Metabolite Interconversion Enzyme	*Hmgcl*	0.00470492	low
	*Nit2*	0.0211131	high
	*Hadha*	0.0337918	low
Nucleic Acid Binding Protein	*SUGP2*	0.019042	low
	*Syncrip*	0.0239965	low
Protein Modifying Enzyme	*PSMD13*	0.0112771	high
	*CAMKV*	0.0247766	low
	*Tpp2*	0.0352463	high
Translational Protein	*EFTUD2*	0.0006042	low
	*Eif3e*	0.0387402	high
Other Protein	*SLC32A1*	0.00091751	high
	*MPDU1*	0.00377415	high
	*Cyb5r3*	0.00814946	high
	*Tmx2*	0.0124046	high
	*Slc2a1*	0.0276784	low

**Table 3 pharmaceuticals-13-00213-t003:** List of 11 retinal proteins which significantly differentially expressed in H_2_S treated group comparing to the CTRL group. Proteins were classified into different categories by PANTHER classification system.

Protein Class	Gene Name	*p*-Value	Abundance
Calcium-binding Protein	*Anxa6*	0.0245515	low
Cytoskeletal Protein	*Dynlrb1*	0.0227117	high
Metabolite Interconversion Enzyme	*Glrx3*	0.0010183	high
	*Ndufa5*	0.0473985	low
Nucleic Acid Binding Protein	*SNRPG*	0.0125193	low
Translational Protein	*Rps28*	0.0406697	low
	*COPS6*	0.0463107	high
Other Proteins	*DDX5*	0.0242271	high
	*KRT1*	0.0326712	low
	*NUP205*	0.0335003	high
	*CXorf22*	0.0419458	low

**Table 4 pharmaceuticals-13-00213-t004:** List of most significantly modulated canonical pathways due to I/R.

	Canonical Pathways	−log(*p*-Value)	Changes	Molecules
Mitochondrial Homeostasis and Function	Ketogenesis	3.62	Downregulated	*HADHA*, *HMGCL*
	Protein Ubiquitination Pathway	1.7	Upregulated	*HSPA1L*, *PSMD13*, *USO1*
	Leucine Degradation I	1.72	Downregulated	*HMGCL*
	Ketolysis	1.68	Downregulated	*HADHA*
Neuronal Calcium Dysregulation	Calcium Transport I	1.68	Upregulated	*ATP2B2*
Cytotoxicity Regulation	Formaldehyde Oxidation II (Glutathione-dependent)	2.37	Downregulated	*ESD*
	Anandamide Degradation	2.2	Upregulated	*FAAH*
	Purine Ribonucleosides Degradation to Ribose-1-phosphate	1.83	Downregulated	*PNP*
Reactive Oxygen Species (ROS) Scavenging	Heme Degradation	2.07	Upregulated	*BLVRA*
	Vitamin-C Transport	1.31	Downregulated	*SLC2A1*
Neural Transduction	Phototransduction Pathway	2.25	Downregulated	*CNGA1*, *GNGT1*
	GABA Receptor Signaling	1.76	Upregulated	*AP2M1*, *SLC32A1*
Vascular Function	eNOS Signaling	1.35	Downregulated	*CNGA1*, *HSPA1L*
	Protein Kinase A Signaling	1.29	Upregulated	*ACP1*, *CAMK2B*, *CNGA1*

I/R vs. CTRL.

**Table 5 pharmaceuticals-13-00213-t005:** Clustered overview of H_2_S regulated canonical pathways. (**A**) The canonical pathways restored by H_2_S. (**B**) The canonical pathways regulated by H_2_S comparing to CTRL.

	Canonical Pathways	−log(*p*-Value)	Changes	Molecules
Mitochondrial Homeostasis and Function	Ketogenesis	3.78	Upregulated	*HADHA*, *HMGCL*
	Leucine Degradation I	1.8	Upregulated	*HMGCL*
	Ketolysis	1.76	Upregulated	*HADHA*
Neural Transduction	GABA Receptor Signaling	1.91	Downregulated	*AP2M1*, *SLC32A1*
ROS Regulation	Vitamin-C Transport	1.38	Upregulated	*SLC2A1*
	HIF1α Signaling	0.742	Upregulated	*SLC2A1*
**Canonical Pathways**		**−log(*p*-Value)**	**Changes**	**Molecules**
NER pathway		1.18	Upregulated	*COPS6*
Oxidative Phosphorylation		1.16	Downregulated	*NDUFA5*
Estrogen Receptor Signaling		1.04	Upregulated	*DDX5*
Corticotropin Releasing Hormone Signaling		1.02	-	*KRT1*
Phagosome Maturation		0.708	-	*DYNLRB1*

(A) H_2_S vs. I/R. (B) H_2_S vs. CTRL.

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
