# Peer review of "Proteomics Reveals the Potential Protective Mechanism of Hydrogen Sulfide on Retinal Ganglion Cells in an Ischemia/Reperfusion Injury Animal Model"

_pharmaceuticals, 2020, doi:10.3390/ph13090213_

Round 1

Reviewer 1 Report

The manuscript entitled “Proteomics reveals the potential protective mechanism of hydrogen sulfide on retinal ganglion cells in an ischemia/reperfusion injury animal model” by Hanhan Liu and co-workwers, explores the potential protective mechanisms activated by hydrogen sulfide in acute glaucoma physiopathology using an animal model. Proteomic results provide evidence for a group of differentially expressed proteins following ischemia and reperfusion, and restored by hydrogen sulfide treatment. The results also indicate the existence of a second group of proteins differentially expressed after treatment with hydrogen sulfide. The authors claim that they provide the first evidence for the complex role of hydrogen sulfide as a protecting agent of retinal ganglion cells against ischemia and reperfusion damage.

 General comments:

The findings reported in this manuscript are interesting and may contribute to elucidate the molecular pathways involved in neuroprotection of retinal ganglion cell by hydrogen sulfide. The authors carried out a relevant effort to interpret and discuss all the proteomic information obtained in the study. However, the work presents important limitations that should be addressed.

Main concerns:

  1. None of the differentially expressed proteins have been confirmed by a second technique. Differential expression of at least selected representative proteins must be confirmed by ELISA or Western blotting. Alternatively, this issue must be discussed in detail as one of the main limitations of the study.
  2. Values of differential expression fold-change must be included in the main tables and discussed properly. Explain whether a fold-change threshold value was set up to identify differentially expressed proteins.
  3. The number of retinas and animals used in each experimental group and in the three replicas must be clearly indicated.

Other concerns:

  1. Abstract. Please state the objective of the study and revise the first sentence.
  2. Revise the second sentence in the Results section (page 2, “Compared to the contralateral control (1310.6 ±236.4 RGC/mm2)…”).
  3. Page 18, paragraph 4.4. The last sentence is incomplete.
  4. Page 18, paragraph 4.7. Explain the meaning of “LFQ”.
  5. Gene symbols must be italicised.
  6. Figure 1. Legend for panels A-C is missing. How may retinas were used in each experimental group for immunohistochemistry? Please, note that the font size is too small and is difficult to read.
  7. Figure 2. Increase the font size, particularly in panels A and C.
  8. Figure 3. Increase the font size. Panels B-E are too small and cannot be seen.
  9. Figure 4. Do the authors mean “retinal proteins” instead of “tear proteins”?
  10. Figure 8. Refer to the different panels in the legend. Explain the meaning of “N” (animals or retinas) and describe how many retinas were used in each replica in panel C.
  11. Figure 8 and page 18. The term “first dimensional gel electrophoresis” is confusing since only one type of electrophoretic separation is used in the study. Please, change “first dimensional gel electrophoresis” to “PAGE”.
  12. Table 1. Use point as a decimal separator.
  13. Table 3. Revise legend and use point as a decimal separator.
  14. Revise the text for missing spaces before and after parenthesis.
  15. Supplementary data 3-5. Explain the meaning of the different parameters, such as “Unique + razor sequence coverage [%]”, “score”, “Intensity”, “Intensity_SHAM_R1”, etc.

Author Response

Dear Reviewer,

First of all, we would like to thank you for reviewing our manuscript and the comments you have risen. Please find attached our response to your comments and a one-to one modification of the manuscript. We sincerely hope we have addressed all those points and concerns you have mentioned. Changes are marked throughout the whole manuscript.

The manuscript entitled “Proteomics reveals the potential protective mechanism of hydrogen sulfide on retinal ganglion cells in an ischemia/reperfusion injury animal model” by Hanhan Liu and co-workwers, explores the potential protective mechanisms activated by hydrogen sulfide in acute glaucoma physiopathology using an animal model. Proteomic results provide evidence for a group of differentially expressed proteins following ischemia and reperfusion, and restored by hydrogen sulfide treatment. The results also indicate the existence of a second group of proteins differentially expressed after treatment with hydrogen sulfide. The authors claim that they provide the first evidence for the complex role of hydrogen sulfide as a protecting agent of retinal ganglion cells against ischemia and reperfusion damage.

 General comments:

The findings reported in this manuscript are interesting and may contribute to elucidate the molecular pathways involved in neuroprotection of retinal ganglion cell by hydrogen sulfide. The authors carried out a relevant effort to interpret and discuss all the proteomic information obtained in the study. However, the work presents important limitations that should be addressed.

Main concerns:

  1. None of the differentially expressed proteins have been confirmed by a second technique. Differential expression of at least selected representative proteins must be confirmed by ELISA or Western blotting. Alternatively, this issue must be discussed in detail as one of the main limitations of the study.

Answer: Thank you for this important remark. However, the exact role H2S plays in neurodegeneration within the retina is so far largely vague. The main focus of this study is to provide a thorough overview of the retina proteome changes related to neuroprotective properties of H2S in retina. Due to the limitation of sample material and the variety of the signaling pathways H2S is involved in, we cannot confirm the differentially expressed proteins by a second technique in the present study design. The findings point to directions for further research. Building on these results, different techniques will be definitely used to confirm the differentially expressed proteins and altered signaling pathways in our future studies.

And we addressed this issue as one of the main limitations of the study in the discussion as reviewer suggested.

  1. Values of differential expression fold-change must be included in the main tables and discussed properly. Explain whether a fold-change threshold value was set up to identify differentially expressed proteins.

Answer: Different statistic approaches are used in proteomics analysis to detect significantly differentially expressed proteins, while the power of the methods can be improved and the true gold standards are yet not known. Instead of applying differential expression fold-change to the proteomics data, we used a type of empirical Bayes method in this study, which is a more powerful analytical approach to proteomics experiments than applying a universal fold-change threshold. This data-driven approach relies on minimal assumptions and presents a stronger protection against false positivity.

The statistical analysis was done in the Perseus software as follow: 1) Data generated from MaxQuant analysis were filtered with a minimum of three valid values in at least one group and the missing values were imputated with a constant using the standard settings in Perseus. 2) all LFQ intensities were subject to a log2 transformation to moderate the variance in each sample. 3) This was followed by a Student’s two-sample t-test for all the groups with p < 0.05 to identify the significantly differentially expressed proteins. 

  1. The number of retinas and animals used in each experimental group and in the three replicas must be clearly indicated.

Answer: Thank you for that important remark. We have addressed that point within the methods section.

Other concerns:

  1. Please state the objective of the study and revise the first sentence.

Answer: Thank you for the comment, changes are made to the abstract accordingly.

  1. Revise the second sentence in the Results section (page 2, “Compared to the contralateral control (1310.6 ±236.4 RGC/mm2)…”).

Answer: Thank you for the comment, the sentence now read “I/R injury resulted in significant reduction in the number of RGC in the operated eyes (1101.7 ±116.4 RGC/mm2) compared to the contralateral control (1310.6 ±236.4 RGC/mm2). While injection of H2S precusor, GYY4137 prior to I/R injury significantly improved RGC survival (1295.4 ±136.1 RGC/mm2).”

  1. Page 18, paragraph 4.4. The last sentence is incomplete.

Answer: Thank you for the comment, the sentence now read”……, vitreous body was then carefully removed.”

  1. Page 18, paragraph 4.7. Explain the meaning of “LFQ”.

Answer: Thank you for the comment, the sentence now read “Briefly, all label-free quantification (LFQ) intensities were subject to a log2 transformation.”

  1. Gene symbols must be italicised.

Answer: Thank you for the comment, all the gene symbols are italicized.

  1. Figure 1. Legend for panels A-C is missing. How may retinas were used in each experimental group for immunohistochemistry? Please, note that the font size is too small and is difficult to read.

Answer: Thank you for the comment. Legend for Figure 1(A-C) is added. And the number of the retina used in each group is clarified, too. Last but not least, font size is increased.

  1. Figure 2. Increase the font size, particularly in panels A and C.

Answer: Thank you for the comment. Font size is increased. Due to the size of the heatmap, font size in panel C is limited, but the detailed list of the proteins can be found in supplementary file 6, and this information is also noted in the figure legend.

  1. Figure 3. Increase the font size. Panels B-E are too small and cannot be seen.

Answer: Thank you for the comment. Font size is increased.

  1. Figure 4. Do the authors mean “retinal proteins” instead of “tear proteins”?

Answer: Thank you for the remark. It is retinal proteins.

  1. Figure 8. Refer to the different panels in the legend. Explain the meaning of “N” (animals or retinas) and describe how many retinas were used in each replica in panel C.

Answer: Thank you for the remark. Figure 8 legend now read “Six retinal protein samples from respective groups were pooled equally into three biological replicates after protein measurements, represented by RI, R2 and R3 (N=3), and subsequently subjected to PAGE.”

  1. Figure 8 and page 18. The term “first dimensional gel electrophoresis” is confusing since only one type of electrophoretic separation is used in the study. Please, change “first dimensional gel electrophoresis” to “PAGE”.

Answer: Thank you for the remark. The term “first dimensional gel electrophoresis” is now changed to “PAGE”

  1. Table 1. Use point as a decimal separator.

Answer: Thank you for the comment, change is made to the table 1.

  1. Table 3. Revise legend and use point as a decimal separator.

Answer: Thank you for the comment, change is made to the table 3.

  1. Revise the text for missing spaces before and after parenthesis.

Answer: Thank you for the comment, changes are made accordingly.

  1. Supplementary data 3-5. Explain the meaning of the different parameters, such as “Unique + razor sequence coverage [%]”, “score”, “Intensity”, “Intensity_SHAM_R1”, etc.

Answer: Thank you for the remark. The meaning of the different parameters in Supplementary data 3-5 is now summarized in Supplementary file 6.

Reviewer 2 Report

This manuscript investigates the potential roles of H2S in glaucoma pathophysiology using the Ischemia-reperfusion model(I/R) in adult Sprague-Dawley rats (n=12) by elevating intraocular pressure to 55mmHg for 60min. Six of the animals received intravitreal injection of H2S precursor prior to the procedure and the retina was harvested 24h later. H2S
significantly improved RGC survival against I/R in vivo (p<0.001).Using label-free mass spectrometry,IPA revealed a significantH2S-mediated activation of pathways related to mitochondrial function, iron homeostasis and vasodilation. The paper is interesting for the role of H2S in protecting RGC from acute iop elevation. However, there are some concerns for this manuscript.

Major concerns:

  1. Animal numbers. Authors described they used 12 SD rats in total. Six of rats received intravitreal injection of H2S precursor prior to the procedure. However, in Fig.1, the counted RGC density, the n=12, is this the number of each group or total? make it clear.
  2.  The author only described the statistical software but didn’t mention the statistical method for comparison of the number of RGC in this study.
  3.  The assumption of the Student’s two-sided t-test is the data which comes from normally distributed population. Originally, the t test was developed for larger sample size and it is no surprise that it does not guarantee reliable results with small sample sizes. The author only used 3 retinal samples per group to perform the proteomic analysis, which is a low sample size to use parametric analysis. There are some good statistical methods such as ROTS, RP, LIMMA, and SAM, which is better choice to analyze the proteomic data for small sample size.  
  4.  In this study, 18 proteins were differentially expressed due to I/R induction and 11 proteins were differentially expressed by H2S treatment. However, the author didn’t confirm these protein levels by another approach. I suggested that the qPCR analysis or western blotting should be used to confirm the proteomic data. This part is lacking. For example: IN 3.1 Changes in Iron Homeostasis and ROS Regulation, all the discussed related protein should be confirmed in the retina, such as ROS, BLVRA and Cyb5r3, etc. So as the discussion of 3.2,3.3, 3.4 and 3.5.
  5. Other experimental evidences  to support that H2S can rescue RGCs in I/R model through the actions of iron regulation, ROS scavenging, the modulation of mitochondrial homeostasis and function, maintenance of retinal
    vascular function and GABA receptor signaling are lacking in the results.
  6. The discussion is lengthy and just based upon the pathway analysis, authors should make it concise since they did not provide ample evidences.
  7.  In figure 1D, the number of spot in the control group, I/R group, and H2S group is plotted about 24, 20, and 15 plots, respectively. But the figure legend described n=12 in each group (?), which is not compatible to the Dot plot. The author should explain the reason specifically in the method section. How many rats used in each group? And how to choose the retina area to count RGC? And how many area are counted per retina?

Minor concerns:

  1. Inappropriate citation of references, P2, references 11-13 are not related to the ocular tissue. Other example is "we documented that GYY4137, a slow-release H2S donor, effectively protected RGC against different glaucomatous injuries in vitro and in vivo, in a dose-dependent manner [6-10]. Only reference 6 is related the text. Authors should check their accuracy of reference citations.
  2. reference 7 and 8 should be chosen as one, but both are not related to the text.

Author Response

Major concerns:

1. Animal numbers. Authors described they used 12 SD rats in total. Six of rats received intravitreal injection of H2S precursor prior to the procedure. However, in Fig.1, the counted RGC density, the n=12, is this the number of each group or total? make it clear.

Answer: Thank you for the remark. Changes are made in the description of the results. In total 12 SD rats were used for this study, all of them received I/R injury and 6 of which received H2S as treated group. 6 of the contralateral eyes were recruited as control group.

2. The author only described the statistical software but didn’t mention the statistical method for comparison of the number of RGC in this study.

Answer: Thank you for the remark. Statistical method is now included.

3.The assumption of the Student’s two-sided t-test is the data which comes from normally distributed population. Originally, the t test was developed for larger sample size and it is no surprise that it does not guarantee reliable results with small sample sizes. The author only used 3 retinal samples per group to perform the proteomic analysis, which is a low sample size to use parametric analysis. There are some good statistical methods such as ROTS, RP, LIMMA, and SAM, which is better choice to analyze the proteomic data for small sample size. 

4.In this study, 18 proteins were differentially expressed due to I/R induction and 11 proteins were differentially expressed by H2S treatment. However, the author didn’t confirm these protein levels by another approach. I suggested that the qPCR analysis or western blotting should be used to confirm the proteomic data. This part is lacking. For example: IN 3.1 Changes in Iron Homeostasis and ROS Regulation, all the discussed related protein should be confirmed in the retina, such as ROS, BLVRA and Cyb5r3, etc. So as the discussion of 3.2,3.3, 3.4 and 3.5.

5.Other experimental evidences  to support that H2S can rescue RGCs in I/R model through the actions of iron regulation, ROS scavenging, the modulation of mitochondrial homeostasis and function, maintenance of retinal
vascular function and GABA receptor signaling are lacking in the results.

6.The discussion is lengthy and just based upon the pathway analysis, authors should make it concise since they did not provide ample evidences.

Answer for 3-6: Thank you for the important comments. The power of the methods we used in this study can be improved and differentially expressed proteins and altered pathways should be confirmed by a second technique. In first part of our study published on IOVS, we discovered the protective effect of H2S in retina against glaucomatous injury. We expected to explore further the undelying mechanisms, however, in the process of the research, we realized that the exact role H2S plays in neurodegeneration within the retina is so far largely vague. Therefore, we conducted the current proteomics study, to provide a thorough overview of the retina proteome changes related to neuroprotective properties of H2S in retina and the findings would point to more specific directions for further research.

But in this study, due to the limitation of sample material and the variety of the signaling pathways H2S is involved in, we cannot confirm the differentially expressed proteins by a second technique in the present study design. Building on these results, different techniques will be definitely used to confirm the differentially expressed proteins and altered signaling pathways in our future studies.

Last but not least we addressed this issue as one of the main limitations of the study in the discussion.

7.  In figure 1D, the number of spot in the control group, I/R group, and H2S group is plotted about 24, 20, and 15 plots, respectively. But the figure legend described n=12 in each group (?), which is not compatible to the Dot plot. The author should explain the reason specifically in the method section. How many rats used in each group? And how to choose the retina area to count RGC? And how many area are counted per retina?

Answer: Thank you for the comments. Quantificantion of RGC is now described in detail in the method section.

Minor concerns:

  1. Inappropriate citation of references, P2, references 11-13 are not related to the ocular tissue. Other example is "we documented that GYY4137, a slow-release H2S donor, effectively protected RGC against different glaucomatous injuries in vitro and in vivo, in a dose-dependent manner [6-10]. Only reference 6 is related the text. Authors should check their accuracy of reference citations.
  2. reference 7 and 8 should be chosen as one, but both are not related to the text.

Answer to 1-2: Thank you for the comments. P2, references 11-13, now 20-22, are not related to the ocular tissue, as H2S has shown various properties in different tissues and systems, however, its exact roles in retina or ocular tissue only became the focus of studies in last few years and remain to be further elucidated. The rest of the references are now re-organized accordingly.

Reviewer 3 Report

The ms entitled “Proteomics reveals the potential protective mechanism of hydrogen sulfide on retinal ganglion cells in an ischemia/reperfusion injury animal model” by Liu et al. deals with the study of protective mechanisms activated by H2S in a glaucoma animal model.

Although the paper is well written and addresses an issue of topical interest, the authors may wish to consider the following prior to publication.

Introduction: the authors mentioned carbon monoxide (CO) nitric oxide (NO) and hydrogen sulfide (H2S), underlining the state-of-the-art of H2S and eye. However, the authors did not mention fundamental papers of this matter. In the background section the authors must be report the relevant papers on this specific field (CO, NO, H2S and eye)

Author Response

Answer: Thank you for the remark. We now included the roles of NO, CO and H2S in relation to the eye in the background section.

Round 2

Reviewer 1 Report

The revised manuscript has been improved. However, some remaining minor issues must be addressed.

  • The objective of the study has not been included in the abstract.
  • The origin of retinas used for immunohistochemistry is still confusing and immunohistochemistry analysis has not been included in the workflow scheme shown in Figure 8. Please clarify these points.
  • Figure 8. The different panels must be specifically described in the legend and they must be cited in the main text.
  • Figure 2. Both heat map scale values and legend are still hard to see.
  • Figure 3. In my opinion, the font size of this figure must be increased.

Reviewer 2 Report

Most of concerned issues have been revised, but in P2 Line 70. Still inappropriate citation. --Alteration of endogenous H2S levels in the retina is also linked to different pathological conditions, and its exogenous donors have been shown to exhibit potential in protecting retinal ganglion cells against insults such as diabetic retinopathy, ischemia–reperfusion injury and N-methyl-D-aspartic acid (NMDA)-induced excitatory neurotoxicity [11-13].- Reference 11-13 are not issue of H2S

Author Response

Thank you for the remark, the citation is now corrected. Reference [11-13] are related to NO, and now [16,19,20] are the correct references to the sentence.